# Geographical Discrimination in Curcuminoids Content of Turmeric Assessed by Rapid UPLC-DAD Validated Analytical Method

**DOI:** 10.3390/molecules24091805

**Published:** 2019-05-10

**Authors:** Amrit Poudel, Jitendra Pandey, Hyeong-Kyu Lee

**Affiliations:** 1Natural Medicine Research Center, Korea Research Institute of Bioscience and Biotechnology, Cheongju-si 28116, Korea; amritpoudel@gmail.com (A.P.); jitupandey01301@gmail.com (J.P.); 2Department of Biodiversity and Bioresources, Satvik Nepal, Dadakonak-27, Pokhara 33700, Nepal

**Keywords:** curcuminoids, turmeric, UPLC-DAD, validation

## Abstract

A fast and reliable ultra-performance liquid chromatography-diode array detection method was developed and validated for the quantitative assessment of turmeric extracts from different geographical locations. Acclaim RSLC PolarAdvantage II column (2.2 μm, 2.1 × 100 mm) was used to analyze individual curcuminoids (curcumin, demethoxycurcumin and bisdemethoxycurcumin) from turmeric samples. The detection was done on ultraviolet absorbance at 425 nm and the column temperature was maintained at 45 °C. A mobile phase consisting of acetonitrile and water was found to be suitable for separation, at a flow rate of 1 mL/min with linear gradient elution. Linearity, specificity, precision, recovery and robustness were measured to validate the method and instrument. Under the described conditions, curcuminoids were collected within one minute. The calibration curve of each curcuminoid showed good linearity (correlation coefficient > 0.999). The relative standard deviations (RSD) of intra-day, inter-day precision and repeatability were less than 0.73%, 2.47% and 2.47%, respectively. In the recovery test, the accuracy ranged from 98.54%-103.91% with RSD values of less than 2.79%. The developed method was used for quantification of individual curcuminoids of turmeric samples. Analysis of turmeric samples from Nepal and South Korea revealed that curcuminoid content was related to geographical location. Turmeric cultivated in warmer climates were found to have higher curcumionoid content than turmeric samples from cooler climates, the southern part of Nepal was found to have two times higher content of curcuminoids than turmeric from the north.

## 1. Introduction

Turmeric is the rhizomatous plant belonging to *Curcuma* genus (Zingiberaceae). The herbaceous perennial *Curcuma* contains approximately 70 species [1]. Most of them are distributed in the tropical and subtropical regions of the world and are widely cultivated in South Asia [2]. In the Indian subcontinent, turmeric is used daily as a cooking spice, beauty aids and a dye. Turmeric has been historically used as herbal medicine in Ayurveda and Traditional Chinese Medicine for different conditions including gastric problems, hepatic disorder, inflammation, cough, infection and dental problems [3]. Turmeric is the primary source of curcuminoids, a yellow colored pigment which is commonly used as a spice and a natural coloring agent. Curcuminoids were reported to have a wide range of pharmacological activities such as anti-inflammatory, anticancer, antioxidant, anti-angiogenic and immunomodulating effects [4]. Some reports showed that curcumin has promising results in the prevention of Alzheimer’s disease [5,6,7].

The major curcuminoids present in turmeric are curcumin, demethoxycurcumin and bisdemethoxycurcumin. The chemical structures of all curcuminoids are similar, though while curcumin contains two methoxy groups at its ortho position, demethoxycurcumin contains only one and bisdemethoxycurcumin contains none. The chemical structures of these curcuminoids are shown in Figure 1. These three curcuminoids are the basis for the quality control of turmeric and turmeric derived herbal formulations [8]. The chemical constituents of any herb depend upon various factors like species, geographical location, climate, parts used [9] and same applies to the content of curcuminoids in any turmeric sample.

There are various spectrophotometric methods to determine the total curcuminoids in any sample but these methods could not quantify individual curcuminoid. TLC (Thin layer chromatography) and HPTLC (High performance thin layer chromatography) provide more qualitative than quantitative determinations and do not provide statistically validated methods for simultaneous determination of curcuminoids in turmeric. Liquid chromatography (LC) coupled with MS/MS or Ultraviolet (UV) detector are the most ideal methods for quantification of curcuminoids. As curcuminoids have good UV absorbance, photodiode array (PDA) detector is more cost effective than mass spectrometer. Recently, an improvement in chromatographic performance has been achieved by the introduction of ultra-performance liquid chromatography (UPLC). UPLC takes full advantage of chromatographic principles to carry out separations with columns of smaller particles. It has also the advantages of high speed, high sensitivity, selectivity as well as specificity when compared with high-performance liquid chromatography (HPLC) [10].

Study on the regional variation of curcuminoids content in *Curcuma longa* has not been done yet, so the present study aims to do a comparative study of curcuminoids content of *C. longa* obtained from different geographical location. For this purpose, a new fast validated analytical method was developed to assess the curcuminoids content of turmeric sample. Moreover, a single step method for the isolation of pure individual curcuminoids has been developed.

## 2. Results

### 2.1. Isolation and Purification of Curcuminoids

Injection of methanol extract in Preparative Liquid Chromatography (PLC) with a gradient solvent system of 40% acetonitrile at 0 min to 80% acetonitrile at 60 min gave three major peaks in UV detection of 425 nm. As there is a possibility of mixing other peaks that are not detected in 425 nm, UV absorbance at 210 nm was taken as secondary detection (Figure 2). Three peaks at 37.5, 42.1 and 46.9 min were collected separately. The purity of isolated compounds was determined by UPLC-MS and were found to be more than 97% pure. The structure determination was done by ^1^H NMR and ^13^C NMR. The compounds collected at 37.5, 42.1 and 46.9 min were found to be curcumin, demethoxycurcumin and bisdemethoxycurcumin, respectively.

### 2.2. Optimization of Chromatographic Condition

The UPLC condition was evaluated for the column, mobile phase, column temperature and flow rate to achieve better chromatographic resolution. For the fast and better separation various columns (Acclaim RSLC PolarAdvantage II, 2.2 μm, 2.1 × 100 mm; Acquity BEH C18, 1.7 μm, 2.1 × 100 mm; Triart C18 ExRS, 1.9 μm, 2.0 × 100 mm; Kinetex C18, 1.7 μm, 2.1 × 100 mm), mobile phase (methanol-water and acetonitrile-water with different modifiers including acetic acid, formic acid and trifluoroacetic acid), column temperature (40, 45 and 50 °C), mobile phase flow rates (0.4, 1 and 1.5 mL/min) and analysis time (2, 5 and 10 min) were examined. The method was optimized with Acclaim RSLC PolarAdvantage II, 2.2μm, 2.1 × 100 mm column and gradient solvent system of acetonitrile and water at a column temperature 45 °C with a flow rate of 1 mL/min. The analysis time was 2 min followed by column washing. The choice of 425 nm as the detection wavelength allowed a high sensibility for all peaks. Figure 3 shows a typical chromatograph of the sample. 

### 2.3. Column Performance

Retention time (RT), theoretical plate count (N), K prime, selectivity (α), tailing factor (TF) and resolution (Rs) were measured to evaluate column performance. Polar advantage II was found to have a short retention time with better resolution and acceptable separation (separation within one minute). Addition of modifier seems not to enhance the separation. The results are summarized in Table 1 and chromatograms are shown in Figure 3.

### 2.4. Method Validation

Analysis for linearity, specificity, precision and accuracy were performed to demonstrate that the method is selective, precise and reproducible. For the linearity, twelve concentrations (125–0.061 μg/mL) of each curcuminoids standard solution were injected individually. Plotting of the peak areas versus concentrations provided the calibration curve equation. As all of the R^2^ values were more than 0.999, linearity was verified. Details about the calibration curve, linear range, LOD and LOQ are shown in Table 2.

Specificity was determined by comparing the peak purity of individual curcuminoids of turmeric extract with standards. Figure 4 indicates that the peaks were pure and lacked interference by impurities. The precision of the developed method was determined by intra-day and inter-day variation, along with repeatability of sextuplets. The RSDs of intra-day and inter-day analysis were in the ranges of 0.07–0.73% and 0.48–2.47%, respectively (Table 3). Similarly, the RSD of repeatability was in the range of 0.26–2.47% (Table 4). These data explain the acceptable degree of precision of the described method.

The accuracy was measured by calculating the recovery of the spiked standard to turmeric extract. In the known concentration of turmeric sample, three concentrations of (low, medium and high) standards were added. The percentage of recovery and RSD were calculated, which showed a recovery of 98.54–103.91% and RSD of 0.07–2.79% (Table 5).

The robustness was determined in order to evaluate the reliability of the established method. The experimental conditions such as analytical instrument, volume of injection, flow rates and column temperature were purposely altered. The retention time (RT), theoretical plate (N), K prime, selectivity (α), tailing factor (TF) and resolution (Rs) were evaluated. The result showed that the analytical factors did not differ greatly depending on instrumental variation (Waters Aquity, Thermo Scientific, Agilent); volume of injection (3, 5, 7 μL); column temperatures (40, 45, 50 °C) and flow rates (0.9, 1, 1.1 mL/min). Chromatograms of turmeric sample injected in three different UPLC instrument are shown in Figure 5 and the results of analytical factors were shown in Appendix A.

### 2.5. Quantification of Turmeric Samples

All turmeric extract samples were quantitatively analyzed for the contents of all three curcuminoids (curcumin, demethoxycurcumin and bisdemethoxycurcumin) using the newly established validated method. Triplicate analysis of each sample was done to determine the mean contents. The results were expressed as the content of each curcuminoid per gram of extract of turmeric sample. *C. longa* from a different geographical location was analyzed for curcuminoids content and curcumin was found to be the primary compound among all curcuminoids. Demethoxycurcumin and bisdemethoxycurcumin were found to be almost equal amount with slight variation while they are found to be approximately half of the curcumin content. *C. longa* extracts from Korea and Nepal from different geographical locations were analyzed to determine regional variation. From Korea, samples from Jeju, Jindo and Koksong were analyzed as they are the representative places for the commercial cultivation of turmeric in Korea. The sample from Jeju was found to have high contents of curcuminoids, which is followed, by Jindo and Koksong (Table 6). Eighteen samples from different locations in Nepal were collected and analyzed to determine the contents of curcuminoids. Samples were found to have a difference in their content as the region of cultivation varied. Curcumin content was found to be high in the samples cultivated in the southern region compared with the northern part of Nepal. A sample from Chitwan was found to have about 165 mg/g of curcumin while the sample from Kalikot was found to have about 79 mg/g of curcumin in the turmeric extract (Table 6, Figure 6).

## 3. Discussion

Different methods to separate curcuminoids from turmeric extract have been used from long back. For the separation of individual curcuminoids (curcumin, demethoxycurcumin and bisdemethoxycurcumin) silica gel column chromatography is generally used [11,12] which is time-consuming and purity of individual curcuminoids is always difficult to achieve. Our method for isolation of individual curcuminoids by PLC is an easy and less time-consuming method. In addition, the optimum purity of each curcuminoids have been obtained. Previously different analytical methods have been developed for qualitative and quantitative determination of curcuminoids [9,13,14,15] with longer analysis time and almost all of the methods used modifiers like formic acid and acetic acid to achieve separation. Our approach provides a very short analysis time where good separation of curcuminoids were achieved within one minute without the use of any modifiers in solvents. The method was validated by various parameters like linearity, specificity, precision and accuracy; all the parameters were within the ranges of ICH and KFDA regulations. Curcuminoids contents were found to be highest in the sample from India which is comparable with contents from Bangladesh, USA and Nepal samples (data shown in Appendix A). These samples cannot represent geographical variations of altitude, season of collection and cultivating temperature within the country. The analysis of samples from different locations within South Korea and Nepal showed enough regional variation within a country. As shown in our data, samples from the northern region in Nepal have lower curcuminoids contents compared with samples from the southern region in Nepal. A similar result is observed in the case of samples from South Korea. In both countries, the southern region is warmer than the northern region. So, among different factors, the temperature could be a critical aspect of curcuminoids content in turmeric samples of different geographical location. Curcuminoid synthase is an enzyme of polyketide synthase type that catalyzes the formation of curcuminoids by condensing p-coumaroyl-CoA and malonyl-CoA [16]. Different types of polyketide synthase are found to be temperature sensitive [17,18]. This could be the probable reason for variation of curcuminoids contents depending on altitude. Further study is required to confirm their interrelationship.

## 4. Materials and Methods

### 4.1. Materials

Methanol used for extraction and UPLC solvents (Methanol and Acetonitrile) were purchased from Burdick & Jackson (Republic of Korea). LC-MS solvents (Acetonitrile for LC-MS, Chromasolv^®^) were bought from Fluka, Analytical. Solvent for NMR (CD_3_OD) was purchased from CellBio (Seongnam-si, Korea). The NMR spectra were carried out on a Varian UNITY 400 FT-NMR spectrometer (Varian, Inc., Palo Alto, CA, USA) using tetramethylsilane as an internal standard. HR-ESI was measured using a Waters Q-Tof Premier spectrometer (Micormass UK limited, Manchester, UK). HPTLC was performed on precoated Kiesel-gel 60 F_254_ (0.25 mm, Merck, Darmstadt, Germany).

### 4.2. Extraction Isolation and Identification of Curcuminoids

Commercially available turmeric powder was extracted with methanol using ultrasonicator. The methanol extract so obtained was then directly injected to PLC (Preparative liquid chromatography) system (Gilson PLC 2020, MI, USA) and the separation was done using Acclaim PolarAdvantage II column (5.0 μm, 250 × 21.2 mm). The mobile phase consisted of acetonitrile and water with gradient elution. The gradient solvent system was optimized as 40% acetonitrile at 0 min to 80% acetonitrile at 60 min with a flow rate of 12 mL/min. UV detection at 425 nm was used as primary detection while 210 nm was used as secondary detection of UV radiation. The structure determination was conducted by proton nuclear magnetic resonance (^1^H-NMR) and carbon13 nuclear magnetic resonance (^13^C-NMR). The values were compared with those from reference [19]. The results were further confirmed with HRMS.

### 4.3. Chromatographic Condition

The chromatography consisted of a UPLC system (Waters Acquity, Milford, MA, USA) with a pump (ACQ-QSM), an autosampler (ACQ-FTN), a column oven and a PDA detector (ACQ-PDA). The Empower 2 software was used to record the output signal of the detector. Chromatographic analysis was carried out using an Acclaim RSLC PolarAdvantage II, 2.2 μm, 2.1 × 100 mm column and the column temperature was maintained at 45 °C. The mobile phase consisted of acetonitrile (A) and Water (B) with linear gradient elution (60%A; 0 min to 80% A; 2 min) at a flow rate of 1 mL/min for better separation. The detection was conducted at 425 nm and the injection volume of each sample was 5 μL.

### 4.4. Column Performance

Column performance was evaluated based on retention time, column efficiency, K prime, selectivity, symmetry factor, and resolution of three peaks of curcuminoids: curcumin, demethoxycurcumin and bisdemethoxycurcumin. All performance parameters were calculated using the US Pharmacopeia [20].

### 4.5. Preparation of Standard and Sample Solutions

The stock solutions were prepared in methanol at a concentration of 2 mg/mL of each curcuminoids. A serial dilution of the stock solution was prepared to establish the calibration curve. Plant samples were obtained from The Foreign Plant Extract Bank (FPEB), The Korea Plant Extract Bank (KPEB) of Korea Research Institute of Bioscience and Biotechnology (Daejeon, South Korea) and The Museum of Natural Drug Resources of Nepal (MNDRN) of Satvik Nepal (Pokhara, Nepal). The respective institute has kept voucher specimens of each sample for reference (Appendix A). Taxonomist Dr. Ramchandra Poudel, Senior Scientific Officer, Nepal Academy of Science and Technology completed the identification of plants from Nepal. Dried plant parts were extracted with methanol using ultrasonication. All methanol extracts were prepared with a concentration of 20 mg/mL in methanol for the first injection. Serial dilution of the sample was carried out when necessary. All solutions were filtered through 0.20 μm filters (PTFE filters, Thermo Scientific, Waltham, MA, USA) before injection.

### 4.6. Method Validation

The method was validated for linearity, limit of detection (LOD) and limit of quantification (LOQ), specificity, precision (inter-day, intra-day and repeatability), accuracy (recovery) and robustness following the international conference on Harmonization (ICH) guidelines and some reports on analysis [21].

### 4.7. Quantification of Turmeric Extractive Solution

The new validated analytical method was applied for the simultaneous determination of individual curcuminoids in different samples of turmeric. The quantification of curcuminoids was done by linear regression of the standards. Each sample was analyzed in triplicate to determine the mean content.

## 5. Conclusions

A simple, less time consuming and single step method for isolation of pure curcuminoids from turmeric extract was developed. The developed analytical method was fast, simple, sensitive, accurate, precise, reproducible and specific for the quantitative determination of curcuminoids in turmeric samples. Analysis of samples cultivated from different geographical locations in Nepal and South Korea showed that variation was dependent on the temperature of the cultivation environment.

## Figures and Tables

**Figure 1 molecules-24-01805-f001:**
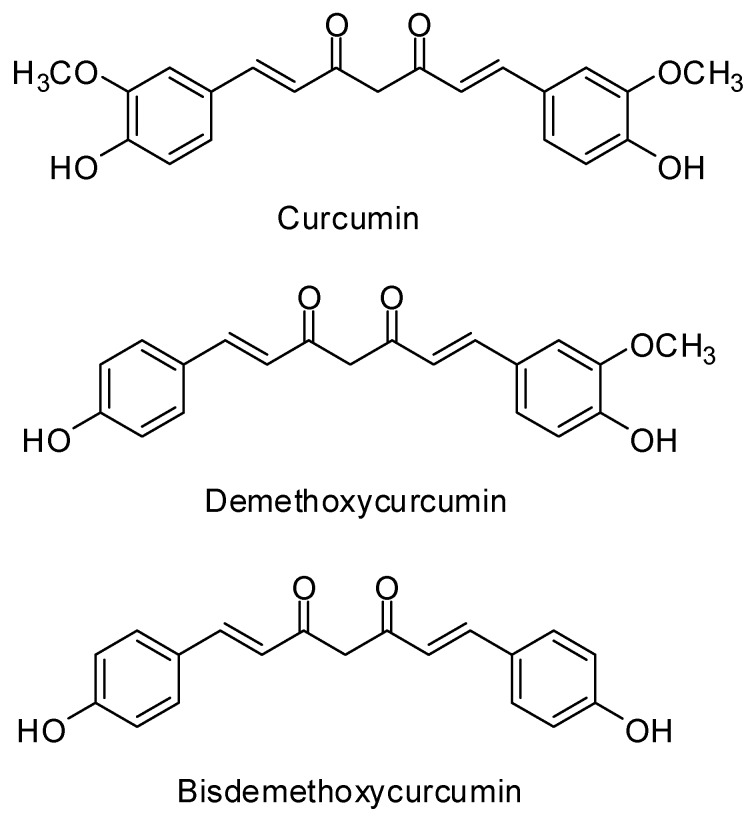
Chemical structures of curcuminoids.

**Figure 2 molecules-24-01805-f002:**
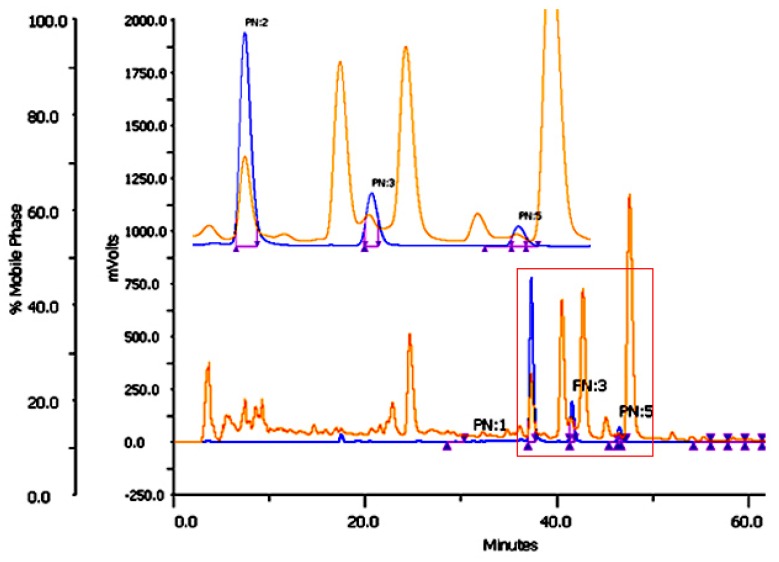
PLC chromatogram for isolation of curcuminoids from methanol extract. PN2: Curcumin, PN3: Demethoxycurcumin, PN5: Bisdemethoxycurcumin.

**Figure 3 molecules-24-01805-f003:**
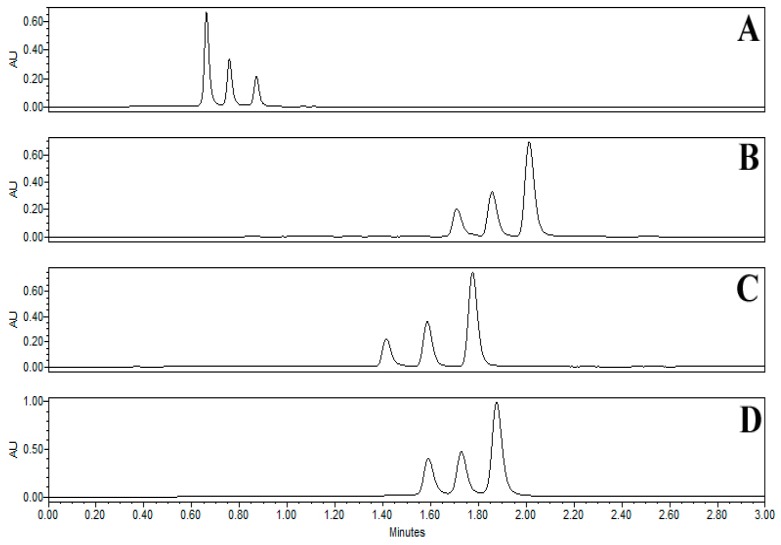
Chromatogram of turmeric sample analyzed in different column at 425 nm UV absorbance. (**A**) PolarAdvantag II; (**B**) Aquity BEH; (**C**) Triart ExRs; (**D**) Kinetex.

**Figure 4 molecules-24-01805-f004:**
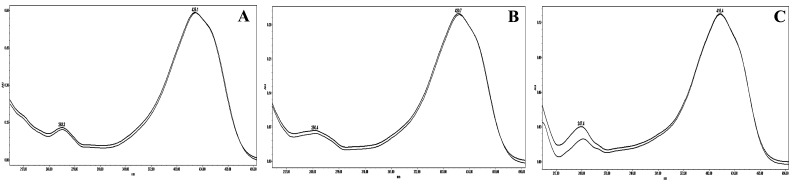
UV spectra of standard curcuminoids (black curve) and turmeric sample (gray curve). (**A**) Curcumin; (**B**) Demethoxycurcumin; (**C**) Bisdemethoxycurcumin.

**Figure 5 molecules-24-01805-f005:**
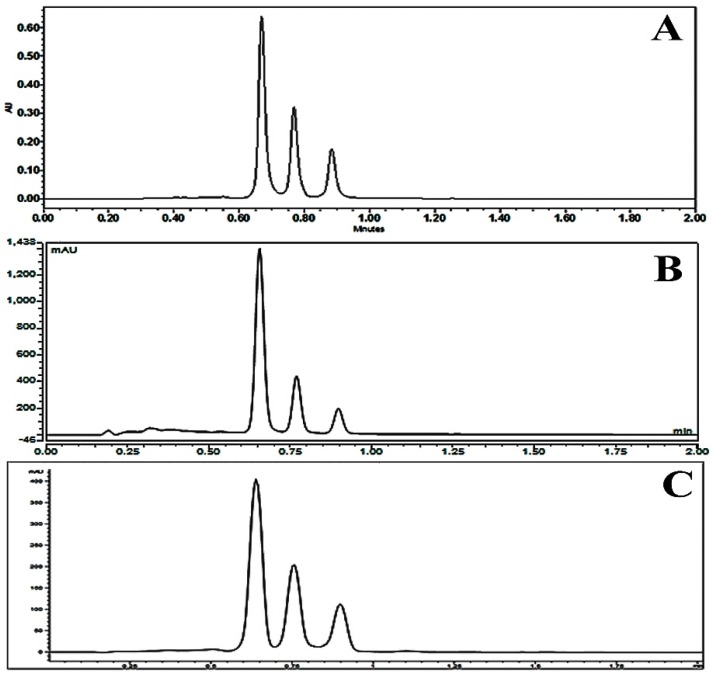
Chromatogram of turmeric sample analyzed in three different instruments. (**A**) Waters Qquity; (**B**) Thermo Scientific; (**C**) Agilent.

**Figure 6 molecules-24-01805-f006:**
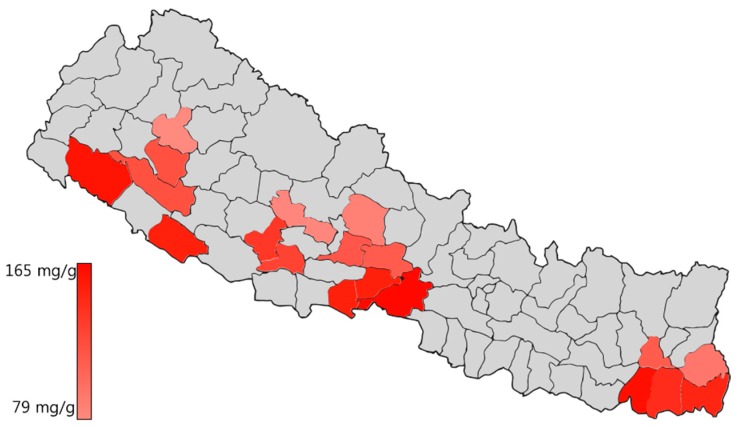
Curcumin content from different geographical locations of Nepal in *C. longa* extract.

**Table 1 molecules-24-01805-t001:** Evaluation of column performance.

Column	Standards	RT	N	K’	α	TF	Rs
Polar Advantage II	Cur	0.701	6476.13	1.93		1.25	
	DMCur	0.800	8260.23	2.27	1.18	1.15	2.81
	BDMCur	0.902	8883.97	2.90	1.27	1.13	3.10
Aquity BEH	Cur	2.012	12227.65	2.32	1.12	1.29	2.11
	DMCur	1.857	10798.41	2.06	1.13	1.30	2.04
	BDMCur	1.709	9397.53	1.82		1.45	
Triart ExRS	Cur	1.775	9911.04	3.15	1.16	1.25	2.63
	DMCur	1.586	8282.66	2.71	1.17	1.27	2.45
	BDMCur	1.414	6965.46	2.30		1.47	
Kinetex	Cur	1.876	9579.02	2.49	1.12	1.22	1.91
	DMCur	1.728	8547.42	2.21	1.13	1.20	1.82
	BDMCur	1.590	7285.05	1.96		1.20	
Modifier							
Without Acid	Cur	0.701	6476.13	1.93		1.25	
	DMCur	0.800	8260.23	2.27	1.18	1.15	2.81
	BDMCur	0.902	8883.97	2.90	1.27	1.13	3.10
Formic acid	Cur	0.682	5349.52	1.88		1.28	
	DMCur	0.786	7304.67	2.29	1.22	1.17	2.62
	BDMCur	0.899	7809.62	2.79	1.21	1.13	2.95
Acetic acid	Cur	0.668	5564.26	1.91		1.21	
	DMCur	0.767	7481.07	2.34	1.23	1.11	2.71
	BDMCur	0.883	8325.92	2.84	1.22	1.05	3.06
Trifluoroacetic acid	Cur	0.660	5583.52	1.84		1.17	
	DMCur	0.751	7472.48	2.24	1.21	1.09	2.55
	BDMCur	0.859	8006.48	2.70	1.21	1.05	2.90

Cur: Curcumin; DMCur: Demethoxycurcumin; BDMCur: Bisdemethoxycurcumin.

**Table 2 molecules-24-01805-t002:** Regression data, LODs and LOQs for curcuminoids analyzed by UPLC *.

Compound	Regression Equation ^†^	R^2^	LOD	LOQ
Curcumin	y = 35831x + 19962	0.999	0.155	0.470
Demethoxycurcumin	y = 36541x + 26194	0.999	0.170	0.517
Bisdemethoxycurcumin	y = 35536x + 13556	0.999	0.190	0.577

* Note: Tests were conducted in triplicates. ^†^ y = peak area; x: concentration (μg/mL).

**Table 3 molecules-24-01805-t003:** Analytical results of intra-day and inter-day variability *.

Standards	Concentration	Intra-Day		Inter-Day	
	(μg/mL)	Mean ± SD (μg/mL)	RSD (%)	Mean ± SD (μg/mL)	RSD (%)
Cur	62.500	63.40 ± 0.10	0.16	63.76 ± 0.31	0.48
	31.250	33.14 ± 0.17	0.51	33.34 ± 0.23	0.68
	15.625	16.29 ± 0.06	0.34	16.46 ± 0.16	0.95
DMCur	31.250	32.41 ± 0.24	0.73	32.50 ± 0.35	1.08
	15.625	17.29 ± 0.01	0.08	17.43 ± 0.17	0.95
	7.813	8.53 ± 0.02	0.26	8.67 ± 0.18	2.13
BDMCur	7.813	7.95 ± 0.01	0.07	8.07 ± 0.11	1.31
	3.906	4.03 ± 0.01	0.17	4.05 ± 0.03	0.68
	1.953	1.85 ± 0.01	0.32	1.88 ± 0.05	2.47

Cur: Curmin, DMCur: Demethoxycurcumi, BDMCur: Bisdemethoxycurcumin; * Note: Tests were conducted in triplicates.

**Table 4 molecules-24-01805-t004:** Repeatability data of curcuminoids in turmeric sample *.

Standards	Amount ± SD (μg/mL)	RSD
Curcumin	30.02 ± 0.12	0.39
Demethoxycurcumin	8.45 ± 0.02	0.26
Bisdemethoxycurcumin	3.14 ± 0.08	2.47

* Note: Tests were conducted in sextuplets.

**Table 5 molecules-24-01805-t005:** Recovery data of spiked standards to turmeric sample *.

Standards	Original (mg/mL)	Spiked (mg/mL)	Found ± SD (mg/mL)	Recovery (%)	RSD (%)
Cur	25.49	7.81	7.82 ± 0.08	100.07	1.02
		3.9	3.97 ± 0.11	101.80	2.79
		1.95	2.02 ± 0.02	103.40	0.81
DMCur	13.17	7.81	7.94 ± 0.01	101.64	0.07
		3.9	3.89 ± 0.00	99.87	0.12
		1.95	1.91 ± 0.01	98.97	0.28
BDMCur	19.08	7.81	7.70 ± 0.09	98.54	1.12
		3.9	4.05 ± 0.08	103.91	2.06
		1.95	1.97 ± 0.01	101.12	0.55

Cur: Curcumin; DMCur: Demethoxucurcumin; BDMCur: Bisdemethoxycurcumin. * Note: Tests were conducted in triplicates.

**Table 6 molecules-24-01805-t006:** Contents of curcuminoids from the different geographical location of Korea and Nepal in *C. longa* extract samples *.

	Longitude(N)/Latitude(E)	Altitude (m)	Cur (mg/g)	DMCur (mg/g)	BDMCur (mg/g)
**Korea Samples**					
Jeju	33.48/126.49	40	48.83 ± 0.12	14.01 ± 0.11	5.53 ± 0.05
Jindo	34.46/126.24	90	37.48 ± 0.34	10.38 ± 0.28	4.25 ± 0.06
Koksong	35.28/127.29	280	35.85 ± 0.05	9.43 ± 0.16	3.73 ± 0.08
**Nepal Samples**					
Chitwan	27.52/84.35	80	165.34 ± 0.08	65.34 ± 0.26	51.54 ± 0.78
Sunsari	26.62/87.18	100	165.12 ± 0.24	86.96 ± 0.20	66.83 ± 0.03
Dhangadi	28.68/80.62	109	159.80 ± 0.02	67.07 ± 0.27	95.33 ± 0.65
Nawalparasi	27.64/83.88	189	150.75 ± 0.69	66.48 ± 0.11	80.58 ± 0.16
Banke	28.14/81.77	215	148.08 ± 0.50	70.31 ± 0.14	65.23 ± 0.16
Jhapa	26.63/87.89	280	142.01 ± 0.37	73.26 ± 0.14	80.41 ± 0.45
Morang	26.67/87.46	400	135.26 ± 0.64	65.95 ± 0.25	86.85 ± 0.19
Pyuthan	28.10/82.85	450	132.21 ± 0.37	54.83 ± 0.26	44.48 ± 0.17
Arghakanchi	27.98/83.03	525	124.93 ± 2.46	59.88 ± 1.60	64.47 ± 2.20
Surkhet	28.51/81.77	750	119.66 ± 0.01	68.52 ± 0.08	51.38 ± 0.06
Dailekh	28.92/81.64	800	119.60 ± 0.07	64.10 ± 0.26	14.44 ± 0.29
Syangja	28.01/83.80	900	116.77 ± 0.10	79.07 ± 0.01	70.35 ± 0.30
Tanahu	27.94/84.22	988	115.39 ± 0.08	62.53 ± 0.06	85.24 ± 0.13
Dhankuta	26.98/87.32	998	114.89 ± 0.25	50.81 ± 0.09	50.52 ± 0.22
Illam	26.87/87.93	1060	103.94 ± 0.06	64.63 ± 0.62	116.09 ± 3.09
Kaski	28.26/84.01	1530	91.57 ± 0.27	63.75 ± 0.53	109.25 ± 0.75
Baglung	28.27/83.58	1750	80.28 ± 0.23	36.92 ± 0.09	43.95 ± 0.19
Kalikot	29.20/81.73	2150	79.14 ± 0.10	30.87 ± 0.11	32.07 ± 0.07

Cur: Curcumin; DMCur: Demethoxucurcumin; BDMCur: Bisdemethoxycurcumin; *Note: Test were conducted in triplicates.

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
