# Peer review of "Geographical Discrimination in Curcuminoids Content of Turmeric Assessed by Rapid UPLC-DAD Validated Analytical Method"

_molecules, 2019, doi:10.3390/molecules24091805_

Round 1

Reviewer 1 Report

The manuscript from Poudel et al. is of scientific interest. However, I would recommend the authors to focus the manuscript on Curcuma content variation from Nepalese samples as the author has collected and analyzed for the different geographical region on Nepal. As the data from other countries are limited. I would suggest to include them in the supplementary information and mention them in the discussion. Additionally, the HPTLC profiling of different Curcuma spp. is also not in the theme of the manuscript, I would recommend focussing on the geographical distribution in the content in the context of Nepal as mention earlier. 

The data (sample) from each country is not enough. So I would request the author to keep the data from figure 7 in supplements and mention as an outlook for future studies and focus this manuscript on the data obtained from Nepal. In the case of Korea, only three samples are taken. India, China, and the USA, and very vast geographical diversity and the data and not enough to represent the country data.

In figure 8, the data on curcumin content from a different region of Nepal can be represented as a heat map or else it is the repetition of table 6. 

I would recommend the authors to include latitude and longitude for the place in table 6

Minor corrections

Title= rephrase it

Line 3 Fast to rapid

Line 21 curcuminoid

Line 37 curcuminoid

Line 28 a cooler climate, the southern

Line 29 the north

Line 36 a cooking spice

Line 38 gastric

Line 39 main to primary

Line 40 a spice

Line 43 the prevention

Line 50 turmeric

Line 51 depend

Line 52 the content

Line 58 font is changed

Line 63 are most to is the most

Line 68 the advantage of

Line 72 a comparative

Line 75 the isolation

Line 75 curcuminoids

Line 84 was to were

Line 91 the column

Line 121 curcuma to Curcuma

Line 132 validation

Line 149 explain the acceptable

Line 192 the results

207 a different country

208 the primary compound

209 the different

216 Bangladesh

223 a difference

231 an approximate

234 the different

267 in both countries

272 the variation

339 each sample

Figure 6c make a monochrome image as A and B

Author Response

We would like to express our humble gratitude to reviewer 1 for precious suggestion and taking time to review our manuscript in detail.

Point 1: The manuscript from Poudel et al. is of scientific interest. However, I would recommend the authors to focus the manuscript on Curcuma content variation from Nepalese samples as the author has collected and analyzed for the different geographical region on Nepal. As the data from other countries are limited. I would suggest to include them in the supplementary information and mention them in the discussion. Additionally, the HPTLC profiling of different Curcuma spp. is also not in the theme of the manuscript, I would recommend focussing on the geographical distribution in the content in the context of Nepal as mention earlier. 

Response 1: We would like to really appreciate reviewer’s suggestion. The data from other countries are included in supplementary information and they are discussed in discussion section. The HPTLC profiling of different Curcuma spp. is removed from the manuscript.

Point 2: The data (sample) from each country is not enough. So I would request the author to keep the data from figure 7 in supplements and mention as an outlook for future studies and focus this manuscript on the data obtained from Nepal. In the case of Korea, only three samples are taken. India, China, and the USA, and very vast geographical diversity and the data and not enough to represent the country data.

Response 2: The figure 7 has been included in supplementary materials. In Korea those three places, jeju, jindo and koksong are major places of commercial turmeric production so these area almost represent the turmeric production in South Korea.

Point 3: In figure 8, the data on curcumin content from a different region of Nepal can be represented as a heat map or else it is the repetition of table 6. 

Response 3: A heat map of curcumin content from different region of Nepal has been included in the manuscript.

Point 4: I would recommend the authors to include latitude and longitude for the place in table 6

Response 4: Information of latitude/longitude along with altitude of place of collection has been included in the table.

Point 5: Minor corrections

Response 5: We would like to thank reviewer for such detail corrections. All the minor corrections have been corrected and changed in manuscript with track change.

Point 6: Figure 6c make a monochrome image as A and B

Response 6: A monochrome image of C as that of A and B has been made and replaced in manuscript.

We would like to thank Reviewer 1 for the suggestions. We thoroughly reviewed the manuscript for any syntax error and English language. All the corrections were made in the manuscript with track change.

Reviewer 2 Report

Opinion related to paper entitled: „Geographical discrimination in curcuminoids content of turmeric assessed by fast UPLC-DAD validated analytical method”.

As I noted in the form, the work is interesting, well made and written in clear language. Modern methods were applied.

The only remark is that section of conclusions have not been sufficiently distinguished.

List of comments:

Line 33 should be belonging to Curcuma genus;

Line 37 should be Traditional Chinese Medicine;

Line 50 should be turmeric, “other” remove “…turmeric and turmeric derived herbal…”;

Line 174 why names of the compounds are written with capital letter, shoul be demethoxycurcumin;

Line 278 should be Chromasolv;

Line 319 give details of filter 0.2 µm;

Author Response

We would like to express our humble gratitude to reviewer 2 for taking time to making precious comments in our manuscript.

Point 1: The only remark is that section of conclusions have not been sufficiently distinguished.

Response 1: The conclusions section has been added in the manuscript (Line 341-346).

Point 2: Line 33 should be belonging to Curcuma genus;

Response 2: Curcuma species is changed to Curcuma genus

Point 3: Line 37 should be Traditional Chinese Medicine;

Response 3: traditional Chinese medicine is changed to Traditional Chinese Medicine

Point 4: Line 50 should be turmeric, “other” remove “…turmeric and turmeric derived herbal…”;

Response 4: trumeric is changed turmeric and ‘other’ is removed.

Point 5: Line 174 why names of the compounds are written with capital letter, shoul be demethoxycurcumin;

Response 5: The name of the compounds are long so to adjust them in the table a short form was used with capital letter. The full name has been mentioned in the footer of the table.

Point 6: Line 278 should be Chromasolv;

Response 6: Chromasov is changed to chromasolv

Point7: Line 319 give details of filter 0.2 µm;

Response 7: The detail of the filter (PTFE filters, Thermo Scientific, USA) is added in the manuscript.

We would like to thank Reviewer 2 for the suggestions. We thoroughly reviewed the manuscript for any syntax error and English language. All the corrections were made in the manuscript with track change.